# Hyperspectral technology and machine learning models to estimate the fruit quality parameters of mango and strawberry crops

Salah Elsayed[1,2], Hoda Gala[3], Mohamed S. Abd El-baki[4], Mohamed Maher[4], Ahmed Elbeltagi[4]*, Ali Salem[5,6]*, Abdallah Elshawadfy Elwakeel[7], Osama Elsherbiny[4], Nadia G. Abd El-Fattah[4]

1 Evaluation of Natural Resources Department, Environmental Studies and Research Institute, Agriculture Engineering, University of Sadat City, Menoufia, Egypt, 2 New Era and Development in Civil Engineering Research Group, Scientific Research Center, Al-Ayen University, Nasiriyah, Thi-Qar, Iraq, 3 Evaluation of Natural Resources Department, Pomology, Environmental Studies and Research Institute, University of Sadat City, Menoufia, Egypt, 4 Faculty of Agriculture, Agricultural Engineering Department, Mansoura University, Mansoura, Egypt, 5 Faculty of Engineering, Civil Engineering Department, Minia University, Minia, Egypt, 6 Faculty of Engineering and Information Technology, Structural Diagnostics and Analysis Research Group, University of Pecs, Pécs, Hungary, 7 Faculty of Agriculture and Natural Resources, Agricultural Engineering Department, Aswan University, Aswan, Egypt

* salem.ali@mik.pte.hu (AS); ahmedelbeltagy81@mans.edu.eg (AE)

**Data Availability Statement:** All data are presented within the article.

**Funding:** The author(s) received no specific funding for this work.

## Abstract

Using chemical laboratory procedures to estimate the fruit quality parameters (biochemical parameters) of mango "Succarri" and strawberry "Florida" as indicators of ripening degrees in a large area presents challenges such as low throughput, labor intensity, time consumption, and the need for multiple samples. So, using spectral reflectance-based proximal remote sensing to quickly and accurately measure biochemical parameters in different fruits is important to find the best time to harvest, make food ripen faster, and the processing of food easier. This has significant economic and ecological advantages. The objective of this study was to evaluate the biochemical parameters of mango and strawberry fruits at various ripening stages. This was done by utilizing a combination of established and newly developed spectral reflectance indices (SRIs) in conjunction with machine learning (ML) models, including artificial neural networks (ANN), random forests (RF), and decision trees (DT). For mango fruit, the parameters estimated were chlorophyll content, total soluble solids (TSS), and firmness, whereas for strawberry fruit, the parameters were L*, b*, TSS, and firmness. These results revealed significant differences in SRI values across various ripening stages, indicating variances in the fruit's biochemical parameters. The newly developed SRIs showed superior efficacy in evaluating these parameters. The integration of SRIs with diverse ML models proved to be a successful strategy for precisely estimating biochemical parameters. For mango's biochemical parameter prediction, the ANN models demonstrated $R^2$ values ranging from 0.92 to 1.00 and from 0.93 to 0.98 for training and testing, respectively. On the other hand, the RF models exhibited $R^2$ values ranging from 0.98 to 1.00 and from 0.93 to 0.99 during training and testing, respectively. The DT models showed high performance, with $R^2$ values ranging from 0.95 to 1.00 and from 0.88 to 0.99 for the training and testing phases. For strawberry's biochemical parameter prediction, the ANN models

**Competing interests:** The authors have declared that no competing interests exist.

achieved R2 values between 0.75 and 0.91 and between 0.58 and 0.91 during training and testing phases, respectively. On the other hand, RF models showed R2 values between 0.85 and 0.91 during training and between 0.74 and 0.86 during testing. The DT models demonstrated excellent results, with R2 values ranging from 0.75 to 0.91 for the training set and 0.74 to 0.81 for the testing set. It can be concluded that combining SRIs with ML models, such as ANN, RF, and DT, can accurately predict the biochemical properties of mango and strawberry fruits.

## 1. Introduction

Mango and strawberry are two of the most preferred fruit crops worldwide [1]. These fruits contain various vitamins such as vitamin B, vitamin C and vitamin E, as well as minerals, phenolic compounds, anthocyanins, flavonoids and other useful nutrients [2,3]. Mango and strawberry can be consumed by population fresh or utilized in the production of juice and marmalade, as well as in the manufacturing of other food products [4]. They could be consumed in local markets or transferred to foreign countries. Therefore, it is necessary to harvest fruits at a suitable ripening stage for each case [5]. During the ripening process, several physiological and chemical changes take place, such as alterations in fruit hardness, total soluble solids, and titratable acidity, as well as variations in fruit color [6]. Fruit firmness and titratable acidity exhibit high levels in immature and mature green fruits and de-crease significantly during the ripening stage [7]. On the other hand, total soluble solids increased as ripening progressed [2]. Fruits are categorized into climacteric and non-climacteric categories based on its respiration patterns, which correlate with ripening [8]. Mango fruits are a prototypical climacteric fruit. During the process of ripening, there is a distinct climacteric peak characterized by a sudden rise in the intensity of respiration, connected to the rapid production of ethylene [3]. Mangoes can be harvested in a mature green stage and then continue to ripen after being harvested [2]. On the other side, strawberries are considered non-climacteric fruits, meaning they have no change in respiration, and have very low levels of ethylene production. Therefore, they should be harvested after ripening [3]. Van de Poel et al. [9] discovered that strawberries can ripen after being detached at the green stage, however, this process significantly impacts fruit quality. Strawberries that were separated from the plant did not grow to the same size, had low levels of malic acid, glucose, fructose, and sucrose, and had fewer volatile compounds. As a result, they had a less intense aroma and an undesirable taste compared to strawberries that ripened on the vine [10].

Harvesting fruit at different levels of ripeness, are detrimental factors that lower the marketability of fruit, reduce their economic value, and increase fruit waste. In the fruit and food industries, it is very important to carefully examine and analyze fruit's quality and then deliver a high-quality product to the customer. Today, scientists are trying to invent non-destructive methods to quickly and accurately determine ripe-ness level in fruits. They are searching for ways that are simple, cheap and accessible [11]. Careful analysis of the quality parameters of the products before their delivery to the market is a highly important principle in the fruit and food industry. Such analysis would increase marketability of the products and help us control product waste. Thus, in the post-harvest stage, in order to raise customer satisfaction and improve product marketability [12]. Mango and strawberry fruits were typically assessed by analyzing their chemical properties with specialized equipment like refractometers and colorimeters [13,14]. Accurately assessing fruit ripeness using traditional methods is challenging

due to the requirement for extensive, representative samples and skilled operators. This makes these methods destructive, uneconomical, and impractical for large-scale use. As a result, research has shifted towards replacing traditional methods with nondestructive, rapid, low-cost, and real-time decision-making alternative methods [15–17]. Thus, nondestructive methods for fruit internal quality are of great value in determining optimal harvest time, ensuring high quality and standard fruits. Through the application of such methods, it is possible to gain a competitive advantage and increase profitability.

In recent years, proximal and remote sensing techniques have become a promising alternative to traditional methods for detecting crucial traits closely associated with mango and strawberry fruit ripeness and determining the best time for harvest [18–22]. These techniques have the potential to rapidly and non-invasively gather various characteristics related to mango and strawberry fruit ripeness for a large number of samples on a large scale [23,24]. Among these techniques is the use of portable VIS/NIR or NIR spectroscopy, equipped with multiple sensors that can detect variations in the optical characteristics of the fruit's surface at different wavelengths [25–28]. Therefore, we could use the spectra emitted by the fruit's surface for the direct or indirect assessment of various traits associated with mango and strawberry ripeness.

As mango and strawberry fruits ripen, their metabolic status undergoes various changes. These changes include alterations in cell wall structure, a decrease in chlorophyll content, and firmness, as well as a gradual increase in TSS. These changes result in significant changes in the spectral reflectance of the fruit across the electromagnetic spectrum [29–31]. People commonly observe changes in the spectral reflectance of ripening grapes in the visible spectrum (VIS, 400–700 nm), as well as the red edge (650–720 nm) and near-infrared (NIR, 700–1200 nm) regions of the spectrum [32]. This suggests that a spectroscopy device that covers only the VIS spectrum, and/or the VIS and NIR spectrums could be useful for detecting changes in the characteristics of fruits during their ripeness. For example, Fatchurrahman et al. [33] found that the spectroscopy of VIS/NIR (400–1000 nm) and NIR (900–1700 nm) was effective in assessing the content of TSS and phenols in goji berries at four different maturity stages.

Generally, the most effective wavelengths within the VIS and NIR spectrum are commonly used in specific equations to create specific spectral reflectance indices (SRIs) such as the anthocyanin index (NAI), normalized chlorophyll index (NCI), normalized difference vegetation index (NDVI), pigment sensitive ripening monitoring index (PRMI), and greenness index (GI) [34–36]. These SRIs are commonly used to monitor changes in biochemical and biophysical attributes during fruit ripeness [37–41]. For example, Elsayed et al. [41] found that the normalized spectral indices HPS 760–730, 760–720 and 686–620 showed the highest coefficients of determination for the SPAD value, chlorophyll a, chlorophyll b and chlorophyll t of mango fruits of the Zabdia cultivar. Additionally, the two-band $SRI_{450,640}$, calculated from the blue and red regions in the VIS spectrum, showed the highest $R^2$ values (0.95 and 0.90) for TSS and firmness, respectively, in ripening banana fruits.

The assessment of fruit quality in crops using SRIs often yields inconsistent outcomes in diverse geographical and environmental conditions. Therefore, there is an ongoing need for refining SRIs to enhance their effectiveness as a rapid and straightforward approach for accurately estimating fruit quality parameters. It is of utmost importance to ascertain the optimal algorithmic formulations for the computation of diverse fruit quality attributes, thereby enhancing the efficacy of remotely acquired data in the evaluation of fruit quality [42,43]. Typically, prior studies have primarily concentrated their efforts on utilizing published SRIs for the assessment of various fruit quality attributes. As far as the authors are aware, only a limited number of inquiries have investigated the simultaneous application of distinct techniques for fruit property assessment. The distinctive advantage of the present investigation lies in the methodology employed for selecting the most suitable SRIs for the evaluation of fruit quality

parameters. In this regard, the utilization of correlogram maps stands out as a noteworthy approach, enhancing the study's capability to identify and employ the most effective SRIs in assessing fruit quality attributes.

Although SRIs offer a straightforward approach to the estimation of biochemical parameters, with the potential to enable the development of a portable and lightweight instrument for the rapid and cost-effective assessment and management of biochemical parameters on a significant scale, it is important to note that each SRI is constrained by a finite set of band combinations. The challenge lies in formulating robust SRIs for the assessment of fruit quality attributes amidst diverse and potentially perplexing conditions. These conditions encompass substantial variations in the dimensions of fruit components and their consequential impact on the saturation level of the quality parameters under scrutiny. As a subset of artificial intelligence (AI), machine learning (ML) has grown quickly in this environment. A huge amount of spectrum data can be used by machine learning (ML) to extract important information for accurate classification and self-prediction [44].

Although multiple linear regression is a widely used technique, its ability to reliably forecast the intended output is limited by the non-linearity of its inputs. Machine learning (ML) models have experienced a surge in popularity in recent years and are now extensively utilized, particularly in domains like smart agriculture and the food business. Scala et al. [45] and Torkashvand et al. [46] found that ML models are effective in contributing to increasing the productivity and quality of fruits. ML models such as artificial neural networks (ANN), random forest (RF), and decision tree (DT) are known for their effectiveness in pattern recognition, prediction and classification of the quality of different fruits [47,48]. These approaches are highly proficient at acquiring and representing intricate, frequently non-linear connections between input and output signals, relying on given patterns [49,50]. it has the ability to learn from example datasets in an iterative manner, even without any prior information about the relationships between the process variables. Prior research has established the dependability and usefulness of these models in forecasting fruit attributes [43,46,51–53]. Based on these factors, our study seeks to employ artificial neural networks, random forest, and decision tree algorithms to analyze and define the quality parameters of strawberry and mango fruits.

In the context of this research study, the overarching objective was to assess the effectiveness of SRIs as non-destructive techniques for estimating the characteristics of mango and strawberry fruits, as well as for detecting the quality parameters of these fruits at various stages of maturity. To achieve this, the study set out to accomplish the following specific goals: (i) Quantify the quality parameters of mango and strawberry fruits at different stages of ripening; (ii) Evaluate the suitability of conventional and newly developed SRIs for quantifying the quality parameters of mango and strawberry fruits; and (iv) Assess the performance of ANN, DT and RF models, which are based on SRIs, in predicting the quality parameters of mango and strawberry fruits.

## 2. Materials and methods

### 2.1. Samples collection and analytical procedures

Mango "Succarri" and strawberry "Florida" fruits were collected from a private orchard placed at Cairo-Alexandria Desert Road. Seventy five samples of mango and forty five plates of strawberry fruits in different ripening stages, from mature green to fully ripening, were selected. The collected fruits were free from diseases, insects, or mechanical damage. Fruit samples were transferred directly to the lab to estimate the following parameters. Fruit samples from each type under three different ripening degrees were were transferred directly to the lab to achieve the physical and chemical analysis, as shown in Fig 1.

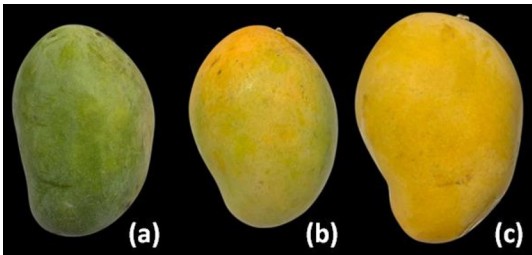 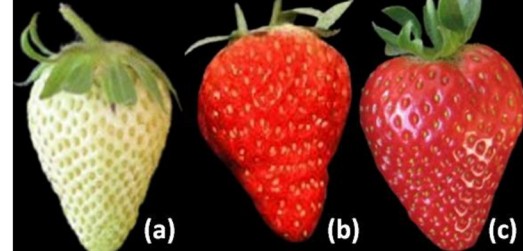

**Fig 1.** Images of mango and strawberry fruits at (a) unripe degree, (b) ripe degree, and (c) overripe degree.

The chlorophyll content or SPAD of mango was measured by using a portable chlorophyll meter, SPAD-502 (Konica-Minolta, Osaka, Japan). The firmness of each fruit for mango and strawberry was measured on different sides using a digital fruit hardness tester (IC-FR5120, China) with a 6 mm probe, and the result was recorded in Newton. Each fruit of mango and strawberry was squeezed separately to measure TSS. TSS was measured with a refractometer (Milwaukee, model MA871, Italy).

Peel color was measured using a precision colorimeter (Sucolor SC-10, China). Measurements were taken at three locations chosen at random for each fruit of strawberry, providing color space parameters of L* and b*. The L* parameter denotes lightness (from 0 for black to 100 for white), and b* is negative for blue and positive for yellow.

## 2.2. Reflectance measurement acquisition and selection of spectral reflectance indices

The spectra of fruit samples for each type of mango and strawberry were collected with a handheld spectrometer (tec5 AG, Oberursel, Germany), which could measure light with a wavelength from 302 nm to 1148 nm with a final spectral interval of 2 nm. On cloud-free days between 11 and 13 h GMT, spectra were acquired from different ripening degrees of fruits under sun radiation. Fruit samples were put above a black sheet, and the spectrometer was held vertically at a nadir position roughly 25 cm above the water surface with a scanning area of 0.03 m$^2$. The final reflection curve for each fruit was the average of the spectral reflectance of the three different places with 10 scans. The reflection was determined by correcting spectrometer results with a calibration factor obtained from a white reference standard (A polytetrafluoroethylene white Spectral on reflectance panel). Processed spectra were then used to derive different SRIs. Table 1 lists some of the most widely used SRIs as well as the method for calculating them, along with references. Sixteen SRIs, including the four most widely used SRIs and twelve newly developed two-band (2-D) SRIs, were illustrated (Table 1). Statistics were displayed on contour maps as determination coefficients ($R^2$) between several measured parameters (SRIs) (Figs 2 and 3). These indices were calculated by integrating potentials at any two wavelengths in a spectrum region ranging from 302 to 1148 nm. Schematic diagram correlating SRIs with four strawberry fruit parameters (L*, b*, TSS, and Firmness) as well as three mango fruit parameters (SPAD, TSS, and Firmness) was shown in Fig 4.

$$RSI = R_1/R_2 \tag{1}$$

$R_1$ and $R_2$ refer to the values of spectral reflectance at various wavelengths.

**Table 1. Published and newly constructed two-band spectral reflectance indices with their formulas and relevant references (Ref.).**

| SRIs | Formula | Ref. |
|---|---|---|
| Published index | | |
| Greenness index (GI) | $R_{554}/R_{677}$ | [54] |
| Pigment-Sensitive Ripening Monitoring Index (PSRMI) | $(R_{750} - R_{678})/R_{550}$ | [55] |
| Anthocyanin index (NAI) | $(R_{760}-R_{720})/(R_{760} + R_{720})$ | [56] |
| Normalized difference vegetation index (NDVI) | $(R_{780} - R_{670})/(R_{780} + R_{670})$ | [57] |
| Newly Ratio Spectral Index (RSI) | | |
| $RSI_{666,636}$ | $R_{666}/R_{636}$ | This study |
| $RSI_{660,620}$ | $R_{660}/R_{620}$ | |
| $RSI_{670,610}$ | $R_{670}/R_{610}$ | |
| $RSI_{618,602}$ | $R_{618}/R_{602}$ | |
| $RSI_{640,590}$ | $R_{640}/R_{590}$ | |
| $RSI_{970,590}$ | $R_{970}/R_{590}$ | |
| $RSI_{764,766}$ | $R_{764}/R_{766}$ | |
| $RSI_{768,770}$ | $R_{768}/R_{770}$ | |
| $RSI_{772,762}$ | $R_{772}/R_{762}$ | |
| $RSI_{766,764}$ | $R_{766}/R_{764}$ | |
| $RSI_{760,858}$ | $R_{760}/R_{858}$ | |
| $RSI_{970,1000}$ | $R_{970}/R_{1000}$ | |

## 2.3. Machine learning methods for mango and strawberry quality assessment

Three machine learning models, like an artificial neural network (ANN), a random forest (RF), and a decision tree (DT), were created using the Python scikit-learn library and Spyder software. These models are used to predict mango and strawberries' biochemical parameters. The input data for these models are spectral indices selected based on its correlation above a predetermined threshold. This study randomly divided the dataset into training (70%) and testing (30%) sets. The random state was determined as 0, which controls randomness and sampling variables to split the nodes. The model predetermines its hyper-parameters before training, instead of learning them from the data. It plays a crucial role in determining the model's performance, as noted by Hossain and Timmer [58]. Cross-validation with a 5-fold size and the grid-search method were employed on the training samples to test different hyperparameter combinations. This testing is employed to select optimal hyperparameters that achieve the best performance for three ML models based on the lowest MSE and the highest $R^2$ value. Schematic diagram of the models was presented in Fig 5.

**2.3.1. Artificial Neural Network (ANN).** ANN models were developed consisting of input layers, hidden layers, and output layers. Each layer consists of neurons. The neuron known as perceptron is similar to multiple linear regression. In this study, stochastic gradient descent (SGD) is selected as the optimization method, as shown in Eq (2). This method is used to adjust the weights and reduce the variation between the expected values and the real laboratory estimated values, as Oymak [59] pointed out. The hyperparameters that need tuning include the number of neurons for each hidden layer (ranging from 2 to 10), the number of hidden layers (varying from 1 to 3),activation function (refer to Table 2 for specific functions used, as described in Sharma *et al.* [60]), a fixed learning rate of 0.001, and maximum iteration (500, 600, 700, 800, 900, 1000). The structure of the ANN model is determined by trial and

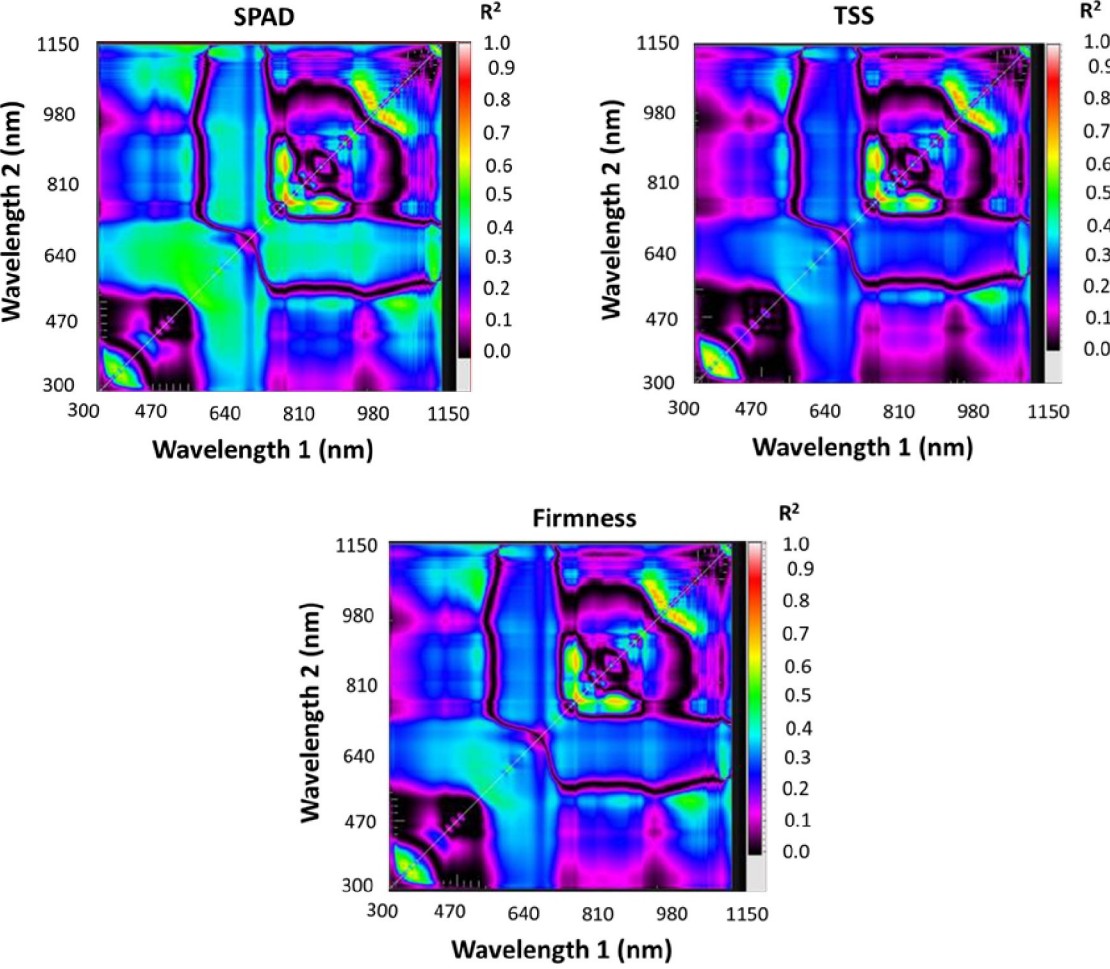

**Fig 2. Correlation matrices displaying (R2) values for possible dull wavelength together ranging from 302 to 1148 nm with SPAD, total soluble solids (TSS), and firmness of mango fruits.**

error, as mentioned by Mijwel [61].

$$\Theta_{j+1} := \Theta_j - \alpha.(Y_a^{(i)} - Y_a^{(i)}).x_j^{(i)} \tag{2}$$

$\Theta_{j+1}$: Weights of the next iteration, $\Theta_j$: Weights of the current iteration, $\alpha$: Learning rate, $x_j^{(i)}$: input feature, $Y_a^{(i)}$: Actual value, $Y_p^{(i)}$: Predict value.

**2.3.2. Random Forest (RF).** Random forest is a widely used machine learning algorithm employed for classification or regression tasks. It utilizes a combination of multiple decision trees to enhance prediction accuracy. At its core, a random forest comprises an ensemble of decision trees generated from random subsets of the available data. In this study, three key hyperparameters were considered: the number of trees in the forest (varying from 1 to 20), the maximum depth of individual trees (varying from 1 to 10), and the criterion function were included were the mean squared error (MSE) and mean absolute error (MAE) methods (refer to Eqs 3 and 4). The primary purpose of employing random forests is to overcome the overfitting problem commonly associated with individual decision trees. By aggregating predictions from multiple trees, the random forest output is evaluated by averaging the results, as suggested by Breiman [62]. This ensemble technique significantly improves the overall

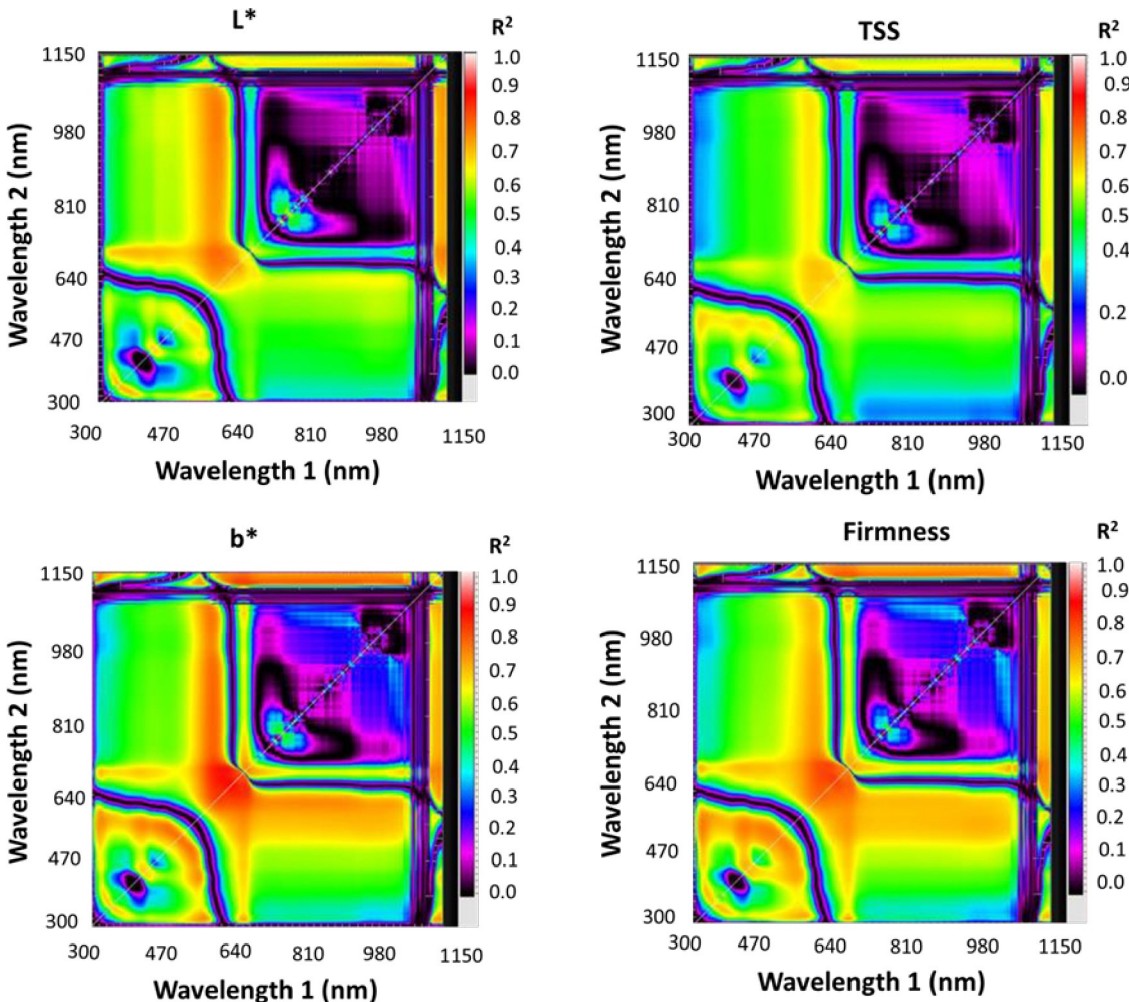

**Fig 3. Correlation matrices displaying (R2) values for possible dull wavelength together ranging from 302 to 1148 nm with total L\*, b\*, total soluble solids (TSS), and firmness of strawberry fruits.**

performance and the model's ability to generalize to unseen data.

$$MSE = \frac{\sum_{i=1}^{N}(Y_a - Y_p)^2}{N} \tag{3}$$

$$MAE = \frac{\sum_{i=1}^{N}|Y_a - Y_p|}{N} \tag{4}$$

Where: $Y_a$ and $Y_p$: represents the actual value and predict value, respectively. N: represent the number of data.

**2.3.3. Decision Tree (DT).** The decision tree algorithm is made up of leaf nodes, decision nodes, branches, and a root node. It is organized like a tree. The root node starts the tree, whereas the decision nodes make decisions that decide the path, which moves from one node to another. The decision nodes end with the leaf nodes, Han et al. [63]. Regression rules are easily created using decision trees. Because the DT doesn't require parameter setting or domain expertise, it is appropriate for exploratory knowledge discovery. During training,

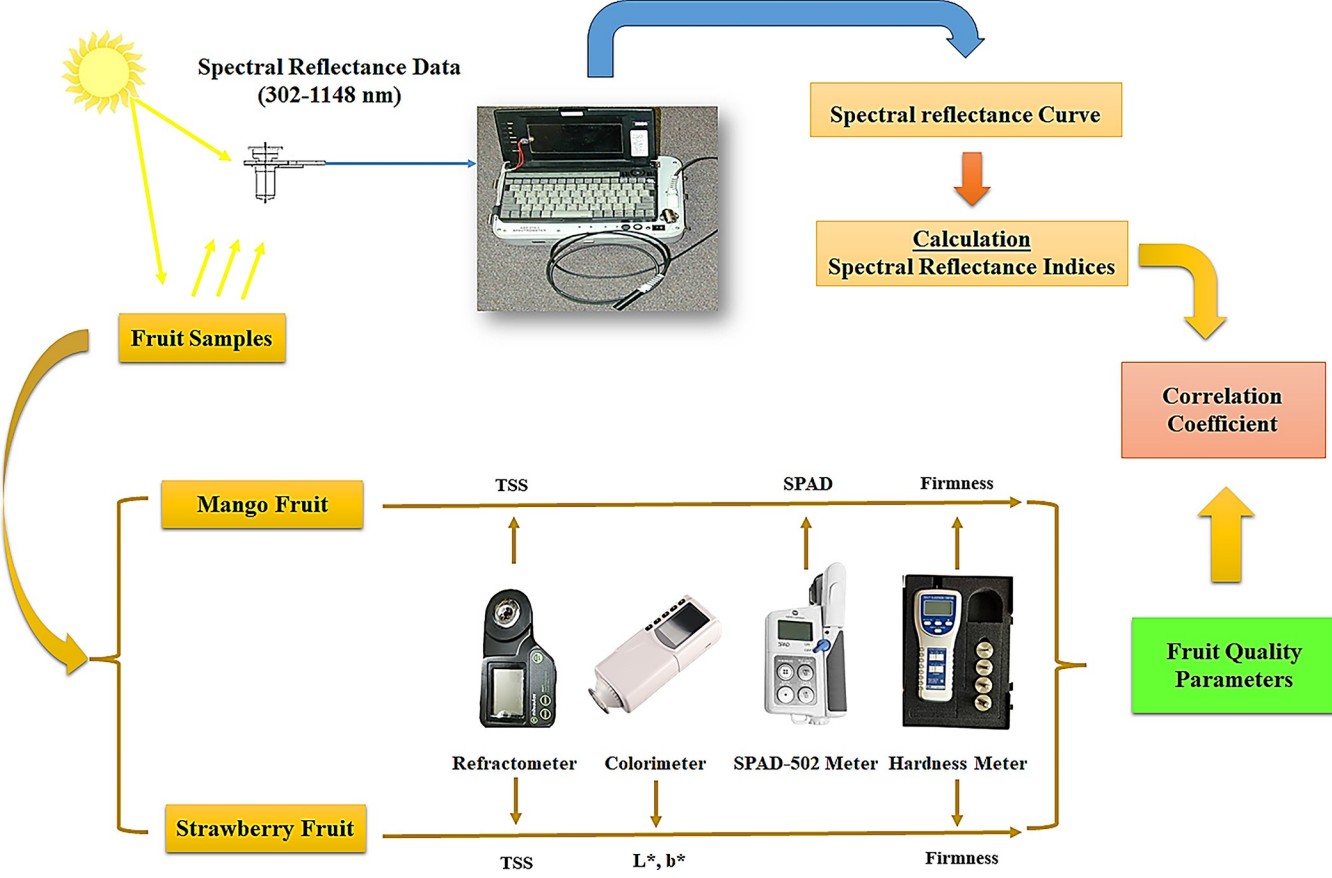

**Fig 4. Schematic diagram correlating SRIs with four strawberry fruit parameters (L\*, b\*, TSS, and Firmness) as well as three mango fruit parameters (SPAD, total soluble solids (TSS), and Firmness).**

hyperparameter optimization was carried out, and the optimal parameters were used to create the top-level model Xia et al. [64]. In this study, two key hyperparameters were taken into account during training to optimize the decision tree model: the maximum depth of the tree (varying from 1 to 10), and the criterion functions used to judge the quality of a split, which were included were the MSE and MAE methods (refer to Eqs 3 and 4).

## 2.4. Models evaluation

In order to assess the efficacy of the three ML models, the MSE (refer to Eq (3)) and the coefficient of determination ($R^2$), as shown in Eq (5) were employed [65,66]

$$R^2 = \frac{\sum \left(Y_a - Y_p\right)^2}{\sum \left(Y_a - Y_{ave}\right)^2} \tag{5}$$

Where: $Y_a$, $Y_p$, and $Y_{ave}$: represents the actual value, predicted value, and average value of original dataset, respectively.

## 2.5. Statistical analysis

Analysis of variance was used to assess data for SPAD, TSS, and firmness of mango fruits, as well L\*, b\*, TSS and firmness of strawberry fruits. Duncan's test was performed to assess the

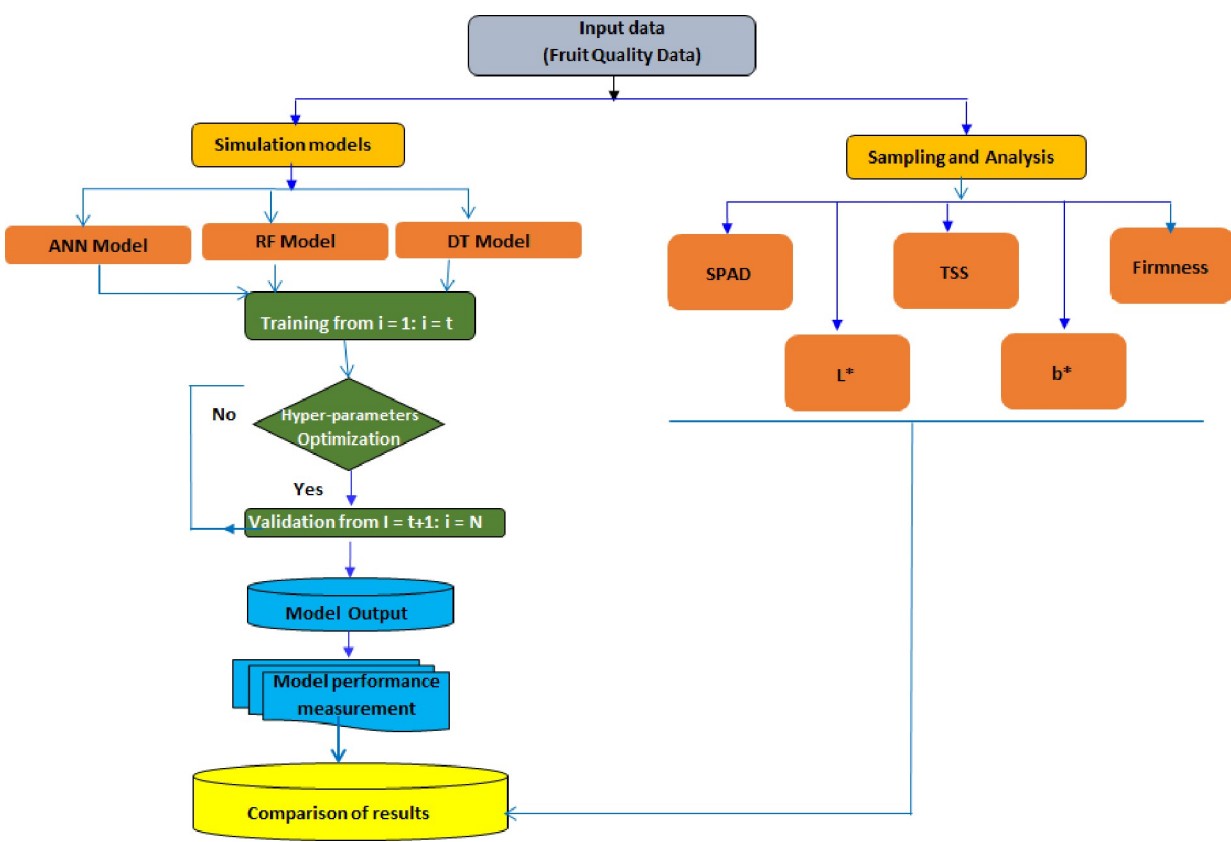

**Fig 5. Schematic diagram for predicting some fruit quality parameters of mango and strawberry using spectral reflectance indices with artificial neural network (ANN), decision tree (DT) and random forest (RF) models.**

differences between the mean values of the ripening stages of the measured parameters at the $P \leq 0.05$ level of probability using SPSS statistical software package version 28.0. The relationship between different measured parameters was determined using Pearson's correlation coefficient matrix. Simple regressions were used to calculate the association between the SRIs and the measured parameters (Excel 2016, v14.0), $R^2$ values and significance levels were determined at 0.001.

## 3. Results and discussion

### 3.1. Effect of ripening stages on quality parameters of mango fruits

The mango fruit quality parameters have a significant impact during the ripening stages. The SPAD values and firmness of the mango fruit exhibited a decline. Conversely, the TSS exhibited a progressive increase in its values across the ripening process. Table 3 displays significant

**Table 2. Types of activation function in ANN models.**

| Name | Equations |
|---|---|
| Hyperbolic Tangent (Tanh) | $f(x) = \frac{(e^x - e^{-x})}{(e^x + e^{-x})}$ |
| Logistic (Sigmoid) | $f(x) = \frac{1}{1 + e^{-x}}$ |
| Rectified Linear Unit (ReLU) | $f(x) = \max(0, x)$ |
| Linear (Identify) | $f(x) = x$ |

**Table 3. Statistical summary of several biochemical parameters of three parameters (SPAD value, TSS, and firmness) of mango fruits across the ripening stages.**

| Mango Fruits | | | | | | | | | | | |
|---|---|---|---|---|---|---|---|---|---|---|---|
| Stage | Unripe | | | | Ripe | | | | Overripe | | | |
| Parameters | Min | Max | Mean | SD | Min | Max | Mean | SD | Min | Max | Mean | SD |
| SPAD | 19.50 | 36.00 | 27.56a | 4.53 | 0.20 | 18.50 | 6.23b | 5.54 | 0.00 | 0.20 | 0.06c | 0.07 |
| TSS (˚Brix) | 8.20 | 15.9 | 11.98c | 2.24 | 16.60 | 19.00 | 17.81b | 0.74 | 17.80 | 21.90 | 19.93a | 1.07 |
| Firmness (N) | 5.90 | 42.5 | 24.68a | 9.50 | 1.30 | 5.85 | 2.24b | 1.29 | 0.00 | 1.25 | 0.78b | 0.39 |

Means having the different alphabetical letter (s) are significantly differ at 0.05 level according to Duncan's multiple range test (P≤ 0.05).

differences in the quality parameters of mango fruit at various ripening stages. The SPAD values vary from 0.00 to 36.00 (dimensionless), the TSS values range from 8.2 to 21.9 (˚Brix), and the firmness values range from 0.00 to 42.5 (N). The reduction in chlorophyll content in fruits can serve as a crucial indicator of fruit quality [67]. According to Medlicott et al. [68], the mango's chlorophyll content decreases as it ripens, as indicated by SPAD measures. The breakdown of chlorophyll pigments and the emergence of other pigments during fruit maturation are responsible for this decrease. Lizada [69] explains that chloroplasts undergo a transformation, containing yellow or red pigments in the process. The unripe stage of mango fruits has high chlorophyll content values, induced by higher SPAD measures (27.56), which then decreases at the ripe stage to 6.23. The chlorophyll content of mango fruit decreases at the overripe stage, which leads to a decrease in SPAD values of 0.06 (Table 3). The gradual decrease in SPAD values corresponds to the color transition of the mango peel from green to yellow or orange, which indicates the ripeness of the fruit. The results of our study are consistent with what Elsayed et al. [41] found in their study. The TSS is an important index for fruit ripening because it indicates the acceptance of nutrients and the economic worth of fruit in trading. Our findings revealed that the overripe stage recorded the highest TSS value (19.93˚-Brix), while the ripe stage recorded 17.81˚Brix. Table 3 displays the lowest TSS value (11.98˚Brix) during the unripe stage. The rise in TSS content can be attributed to the enzymatic transformation of intricate carbohydrates into more basic sugars, aided by the phosphorylase enzyme throughout the maturation procedure, as corroborated by Palafox-Carlos et al. [70]. According to Appiah et al. [71], the increase in TSS during the ripening phase validates the anticipated sweetening effect. Our results support the rising trend of TSS during mango ripening, which is consistent with earlier research by Subedi et al. [72], Islam et al. [73] and Nambi et al. [74]. Meanwhile, the firmness of the fruit is commonly used as an indicator of fruit quality. Our investigation's results revealed that the highest firmness value was achieved at the unripe stage, reaching 24.68 N. At the ripe stage, we observed a clear decrease in firmness value with a 2.24 N. The firmness saw a drop to its lowest value during the overripe stage, with a value of 0.78 N (Table 3). The gradual decrease in firmness can be attributed to the release of water moving from the peel to the pulp. Furthermore, enzymatic activity enhances the degradation of cell wall polysaccharides and the transformation of starch into more basic compounds during fruit ripening [75]. Our findings agree with the observations made by Robles-Sánchez et al. [76] and Palafox-Carlos et al. [70].

## 3.2 Effect of ripening stages on quality parameters of strawberry fruits

The quality parameters of strawberry fruits are clearly affected during fruit ripening. L*, b*, and the firmness of the fruits showed a constant decrease. In contrast, TSS showed a continuous increase during strawberry fruit ripening. Table 4 shows the significant differences between the fruit quality parameters' values at different maturity stages. There is also a wide

**Table 4. Statistical summary of several biochemical parameters of four parameters (L\*, b\*, TSS, and firmness) of strawberry fruits across the ripening stages.**

| | Strawberry Fruits | | | | | | | | | | | |
|---|---|---|---|---|---|---|---|---|---|---|---|---|
| Stage | Unripe | | | | Ripe | | | | Overripe | | | |
| Parameters | Min | Max | Mean | SD | Min | Max | Mean | SD | Min | Max | Mean | SD |
| L* | 35.20 | 43.32 | 39.62a | 2.18 | 31.44 | 42.24 | 35.43b | 2.61 | 29.84 | 33.47 | 31.35c | 1.04 |
| B* | 22.19 | 26.92 | 24.46a | 1.42 | 19.38 | 24.87 | 21.80b | 1.84 | 13.21 | 17.83 | 15.52c | 1.44 |
| TSS (°Brix) | 7.50 | 9.70 | 8.53c | 0.47 | 8.50 | 10.00 | 9.33b | 0.51 | 9.70 | 12.10 | 10.42a | 0.59 |
| Firmness (N) | 5.63 | 7.92 | 7.09a | 0.69 | 4.47 | 6.97 | 5.92b | 0.69 | 3.28 | 4.53 | 3.84c | 0.36 |

Means having the different alphabetical letter (s) are significantly differ at 0.05 level according to Duncan's multiple range test (P≤ 0.05).

range of values for all quality parameters at different fruit ripening stages, as shown in Table 4. The L* value ranged from 29.84 to 43.32, the b* values ranged from 13.21 to 26.92, the TSS value ranged from 7.50 to 12.10, and the firmness value ranged from 3.28 to 7.92. Table 4 displays the findings of our study, which indicate that the maximum L* value (39.62) was found during the unripe stage, followed by 35.43 during the ripe stage, and the lowest value (31.35) was found during the overripe stage. These findings indicate that strawberry fruit's brightness diminishes as it ripens. In other words, as fruit ripens, it becomes darker. These results are consistent with the results of Saad et al. [14]. The highest b* value (24.46) was recorded during the unripe stage, followed by 21.80 during the ripe stage. The lowest b* values (21.80) were observed during the overripe stage (Table 4). The gradual decrease in b* values during ripening processes is attributed to the fact that the fruit turns red as a result of the synthesis of pigments such as carotenoids, anthocyanins, and flavonoids, according to Winarno [77]. This change in color is used as an indicator of the ripening process of strawberry fruits, as pointed out by Cahyono [78]. These results are in agreement with Janurianti et al. [79] and Tsormpatsidis et al. [80]. TSS is an important indicator to determine fruit maturity. In our study, we found significant differences in TSS values during different ripening stages. The highest TSS value is 10.42°Brix during the overripe phase. During the ripe stage, the value was detected as 9.33°Brix. The lowest value of 8.53°Brix was observed during the unripe stage (Table 4). This gradual increase during the ripening processes of strawberry fruit can be attributed to the increased concentration of sugars and organic acids [71,72]. In addition, Ullah [81] and Bhatti [82] also pointed out that with the progress of fruit ripening, the sugar content increases as a result of the hydrolysis of sucrose, which leads to an increase in TSS. With the transition of the fruit from the unripe stage to the overripe stage, the moisture content in the strawberry fruit decreases and, accordingly, the concentration of Tess becomes higher and the increase appears, as Martínez et al. [83] mentioned. The gradual increase is consistent with previous studies by Rahman et al. [84], and Janurianti et al. [79]. The firmness parameter plays an important role in determining the suitability of the fruit for commercialization. It is an indicator of mechanical resistance Gunness et al. [85]. We discovered significant differences between the values of firmness at different ripening stages. During the unripe stage, the highest firmness value is 7.09 N. The ripening stage has a value of 5.92 N. The lowest values of 3.84 N are recorded during the overripe stage (Table 4). The tendency of this parameter to decrease during the ripening process of strawberry fruit corresponds to de Jesús Ornelas-Paz et al. [86].

## 3.3 Variation of SRIs at different ripening degrees for mango and strawberry fruits

The spectral reflectance of fruit surfaces can reveal important information about alterations in the fruit's biochemical parameters. These alterations result in variations in spectral reflectance

**Table 5. Statistical summary of several spectral reflectance indices for mango fruits across the ripening stages.**

| Stage | Unripe | | | | Ripe | | | | Overripe | | | |
|---|---|---|---|---|---|---|---|---|---|---|---|---|
| Parameters | Min | Max | Mean | SD | Min | Max | Mean | SD | Min | Max | Mean | SD |
| $RSI_{764,766}$ | 1.001 | 1.009 | 1.004a | 0.002 | 0.997 | 1.000 | 0.998b | 0.001 | 0.992 | 0.997 | 0.995c | 0.001 |
| $RSI_{768,770}$ | 0.997 | 1.003 | 1.000a | 0.002 | 0.994 | 0.997 | 0.996b | 0.001 | 0.991 | 0.995 | 0.993c | 0.001 |
| $RSI_{772,762}$ | 0.974 | 1.004 | 0.990c | 0.008 | 1.004 | 1.018 | 1.012b | 0.004 | 1.017 | 1.039 | 1.026a | 0.006 |
| $RSI_{766,764}$ | 0.981 | 1.010 | 0.997c | 0.008 | 1.011 | 1.024 | 1.018b | 0.004 | 1.024 | 1.046 | 1.032a | 0.006 |
| $RSI_{760,858}$ | 1.101 | 1.189 | 1.146a | 0.021 | 1.076 | 1.141 | 1.100b | 0.016 | 1.035 | 1.111 | 1.073c | 0.021 |
| $RSI_{970,1000}$ | 0.927 | 0.980 | 0.947a | 0.014 | 0.898 | 0.942 | 0.922b | 0.012 | 0.904 | 0.931 | 0.914c | 0.007 |
| GI | 1.262 | 3.141 | 2.526a | 0.458 | 0.890 | 3.205 | 1.990b | 0.667 | 0.500 | 2.818 | 1.120c | 0.669 |
| PRMI | 0.874 | 1.671 | 1.349a | 0.205 | 0.431 | 1.625 | 1.124b | 0.301 | -0.067 | 1.394 | 0.563c | 0.472 |
| NAI | -0.010 | 0.106 | 0.060a | 0.034 | -0.014 | 0.115 | 0.031b | 0.034 | -0.022 | 0.090 | 0.003c | 0.024 |
| $NDVI_{780,670}$ | 0.367 | 0.713 | 0.624a | 0.086 | 0.179 | 0.716 | 0.514b | 0.154 | -0.001 | 0.662 | 0.256c | 0.217 |

Means having the same alphabetical letter (s) are not significantly differ at 0.05 level according to Duncan's multiple range test.

indices (SRIs) across specific wavelengths, according to Sirisomboon [30]. The biochemical parameters of strawberry and mango fruit are marked by SRIs in Tables 5 and 6, respectively. These SRIs values show significant changes across different ripening stages, attributed to significant variations in biochemical parameters. For mango fruit, quantitative analysis shows a significant variation in the mean values of mango fruit's biochemical parameters. The SPAD value ranged from 0.00 to 36.00, the TSS values ranged from 8.20 to 21.90, and the firmness value ranged from 0.00 to 42.50. These alterations are followed by corresponding changes in the mean values of indices, including $RSI_{764,766}$, $RSI_{768,770}$, $RSI_{772,762}$, and $RSI_{766,764}$, with values ranging from 0.992 to 1.009, 0.991 to 1.003, 0.974 to 1.039, and 0.981 to 1.024, as indicated in Table 5. All SRIs values for mango fruit have a significant difference across various ripening stages. For strawberry fruits, quantitative analysis reveals significant variation in quality parameters. These parameters such as, L*, b*, TSS, and firmness value ranged from 29.84 to 43.32, 13.21 to 26.92, 7.50 to 12.10, and 3.28 to 7.92, respectively. These changes are attributed to alterations in the mean values of indices, namely $RSI_{666,636}$, $RSI_{660,620}$, $RSI_{670,610}$, and $RSI_{618,602}$, with values ranging from 0.715 to 1.328, 0.779 to 1.955, 0.665 to 2.694, and 1.005 to 1.740, as shown in Table 6. Most of the SRIs values for strawberry fruit's quality parameters

**Table 6. Statistical summary of several spectral reflectance indices for strawberry fruits across the ripening stages.**

| Stage | Unripe | | | | Ripe | | | | Overripe | | | |
|---|---|---|---|---|---|---|---|---|---|---|---|---|
| Parameters | Min | Max | Mean | SD | Min | Max | Mean | SD | Min | Max | Mean | SD |
| $RSI_{666,636}$ | 0.715 | 0.877 | 0.77c | 0.040 | 0.759 | 0.887 | 0.84b | 0.036 | 1.062 | 1.328 | 1.17a | 0.075 |
| $RSI_{660,620}$ | 0.799 | 1.088 | 0.88c | 0.071 | 0.901 | 1.117 | 1.01b | 0.056 | 1.373 | 1.955 | 1.62a | 0.143 |
| $RSI_{670,610}$ | 0.665 | 1.092 | 0.78c | 0.106 | 0.793 | 1.121 | 0.95b | 0.087 | 1.608 | 2.694 | 2.04a | 0.268 |
| $RSI_{618,602}$ | 1.005 | 1.271 | 1.08c | 0.070 | 1.086 | 1.293 | 1.17b | 0.053 | 1.457 | 1.740 | 1.57a | 0.075 |
| $RSI_{640,590}$ | 0.998 | 1.948 | 1.26c | 0.245 | 1.286 | 2.079 | 1.59b | 0.204 | 2.892 | 4.689 | 3.57a | 0.483 |
| $RSI_{970,590}$ | 0.801 | 2.093 | 1.29b | 0.322 | 1.406 | 2.330 | 1.68b | 0.247 | 2.379 | 5.667 | 3.93a | 0.832 |
| GI | 0.294 | 1.065 | 0.68a | 0.187 | 0.234 | 0.628 | 0.39b | 0.119 | 0.072 | 0.142 | 0.10c | 0.019 |
| PRMI | 1.075 | 3.952 | 1.97b | 0.809 | 1.617 | 4.285 | 2.74b | 0.719 | 3.618 | 11.919 | 6.56a | 2.015 |
| NAI | 0.003 | 0.294 | 0.016a | 0.007 | 0.011 | 0.025 | 0.019a | 0.004 | 0.006 | 0.028 | 0.016a | 0.006 |
| $NDVI_{780,670}$ | 0.029 | 0.445 | 0.40a | 0.041 | 0.317 | 0.422 | 0.36b | 0.027 | 0.183 | 0.337 | 0.28c | 0.047 |

Means having the same alphabetical letter (s) are not significantly differ at 0.05 level according to Duncan's multiple range test.

show a significant difference during different ripening stages. However, the two indices (RSI$_{970,590}$ and PRMI) show no significant difference between the unripe and ripe stages. The NAI values also show no significant difference between the three ripening stages. In general, the gradual increase or decrease in SRIs values during different ripening stages can be attributed to changes in the quality parameters of fruits Elsayed et al. [67].

## 3.4 Evaluation of spectral reflectance indices (SRIs) to assess the quality parameters

The effectiveness of both previously published and newly devised SRIs in detecting biochemical parameters in mango and strawberry fruits was assessed in this study. Fig 6A and 6B exhibit the findings, illustrating that the newly developed SRIs are superior in assessing biochemical parameters. Nevertheless, the majority of published SRIs also demonstrated correlations with biochemical parameters. For mango fruit, the published SRIs showed $R^2$ values ranging from 0.29 to 0.38 for SPAD, 0.30 to 0.41 for TSS, and 0.22 to 0.31 for firmness. The $R^2$ values exhibited relatively low values for all mango quality parameters when employing the published SRIs, as depicted in Fig 6A. The newly developed SRIs obtained $R^2$ values ranging from 0.71 to 0.91 for SPAD, 0.62 to 0.88 for TSS, and 0.65 to 0.84 for firmness. The mango fruit's biochemical parameters had the highest $R^2$ values with newly developed SRIs, which were derived from the near-infrared (NIR) region. For example, the RSI$_{764,766}$ and RSI$_{768,770}$ displayed $R^2$ values of 0.91 and 0.91 for SPAD, 0.88 and 0.87 for TSS, and 0.84 and 0.85 for firmness, respectively, as seen in Fig 4A. For strawberry fruit, the published SRIs presented $R^2$ values ranging from 0.02 to 0.70 for L*, 0.00 to 0.66 for b*, 0.00 to 0.60 for TSS, and 0.00 to 0.69 for firmness. The $R^2$ values were low with all quality parameters when using the published SRIs, particularly with the NAI index (Fig 6B). Conversely, the newly developed SRIs yielded $R^2$ values ranging from 0.60 to 0.67, 0.71 to 0.84, 0.57 to 0.65, and 0.69 to 0.79 for L*, b*, TSS, and firmness, respectively. The newly developed SRIs, extracted from the red region in the visible spectrum, showed the best $R^2$ values with the biochemical parameters of strawberry fruit. For example, RSI$_{666,636}$ and RSI$_{660,620}$ showed $R^2$ values of 0.67 and 0.66 for L*, 0.84 and 0.83 for b*, 0.63 and 0.63 for TSS, and 0.79 and 0.78 for firmness, respectively, as shown in Fig 6B.

Several studies have highlighted the potential of using spectral reflectance indices to remotely and non-destructively assess different biochemical parameters in fruit [5,79,80,87]. For instance, Elsayed et al. [31] reported that the SRIs produced from visible or near-infrared regions are usefull in determining the soluble solids content and SPAD value of mango fruits. Tijskens et al. [88] and Wei et al. [89] revealed that variations in pigment components for

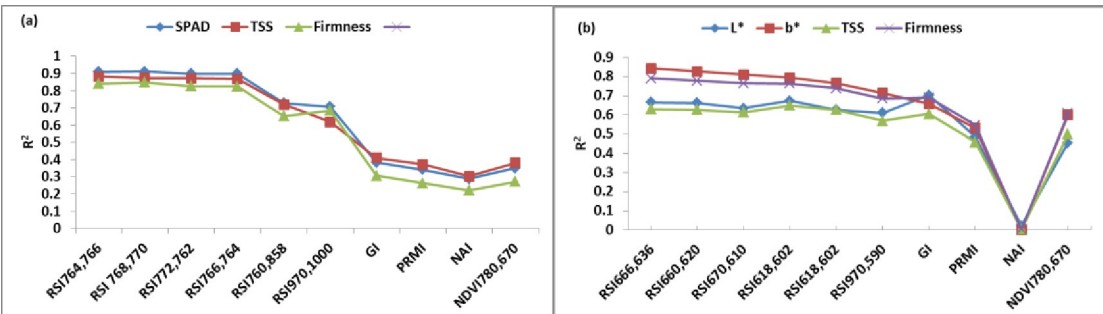

**Fig 6.** Coefficient of determination values of linear regression models of (a) three mango fruit attributes as SPAD, total soluble solids (TSS), and firmness with ten spectral reflectance indices (SRIs) as well as (b) four strawberry fruit attributes as L*, b*, total soluble solids (TSS) and firmness with ten spectral reflectance indices.

strawberry fruit like anthocyanin and chlorophyll could be linked to variations in spectral density in the 550–900 nm region. Furthermore, Yahaya et al. [90] reported that the red wavelengths (600–700 nm) are efficiently absorbed by mango pigments. These wavelengths and the assessments of mango fruit hardness, acidity, and soluble solids content (SSC) showed the strongest relationships, as a consequence. These studies demonstrate how spectral reflectance indices can be used to quickly and non-destructively evaluate fruit quality by estimating and evaluating a variety of biochemical characteristics.

## 3.5. ML-models performance for mango quality assessment

Three machine learning (ML) models have demonstrated remarkable accuracy in predicting important parameters for mango, namely TSS, SPAD, and firmness, as illustrated in Table 7 across both the training and testing phases. The findings of the testing phase are displayed in Fig 7. The artificial neural network (ANN-TSS) model obtained an $R^2$ value of 0.92 and 0.93 for TSS prediction, with corresponding MSE values for the training and testing phases of 1.13 and 0.52, respectively. The ANN-TSS model was built using a single hidden layer that had five neurons (Fig 8A). This model also employed the logistic activation function and underwent 500 iterations. The random forest (RF-TSS) model surpassed even greater accuracy, achieving an $R^2$ value of 0.98 and 0.93, along with an MSE of 0.30 and 0.51 for the training and testing phases, respectively. The RF-TSS model consisted of 15 trees and a maximum depth of 4. This model used the squared error as the criterion function. Moreover, the decision tree (DT-TSS) model performed admirably, showing $R^2$ values of 0.95 and 0.88 for the training and testing phases, respectively, along with MSE values of 0.70 and 0.87. The DT-TSS model had a maximum depth of 3. This model used the squared error as the criterion function. These results demonstrate how well these ML models can predict mango fruit's TSS. In terms of SPAD prediction, the artificial neural network (ANN-SPAD) model performed exceptionally well, with $R^2$ values of 0.99 and 0.98 and MSE values of 0.83 and 1.34 for the training and testing phases, respectively. The ANN-SPAD model's construction had five neurons in each of the two hidden layers, as depicted in Fig 8B. This model went through 700 iterations using the relu activation function. Additionally, the random forest (RF-SPAD) model showed impressive accuracy, giving MSE values of 1.53 and 0.25 and an $R^2$ value of 0.99 and 0.98 for the testing and training phases, respectively. Comprising 20 trees with a maximum depth of 5, the RF-SPAD model

**Table 7. Performance of different ML models incorporate distinct spectral reflectance indices (SRIs) to predict of total soluble solids) TSS(, SPAD, and firmness of Mango fruits after training and testing phase.**

| Target Variables | Models | Optimal Input Variables | Training | | Testing | |
|---|---|---|---|---|---|---|
| | | | $R^2$ | MSE | $R^2$ | MSE |
| TSS | ANN-TSS | RSI764,766—RSI768,770—RSI772,762—RSI766,764—RSI760,858—RSI$_{970,1000}$ | 0.9242 | 1.13 | 0.9288 | 0.52 |
| | RF-TSS | RSI764,766—RSI768,770—RSI772,762—RSI766,764—RSI760,858—RSI970,1000 –GI–PRMI–NDVI$_{780,\ 670}$ | 0.9800 | 0.30 | 0.9300 | 0.51 |
| | DT-TSS | RSI764,766—RSI768,770—RSI772,762—RSI766,764—RSI$_{760,858}$ | 0.9531 | 0.70 | 0.8819 | 0.87 |
| SPAD | ANN-SPAD | RSI764,766—RSI768,770—RSI772,762—RSI$_{766,764}$ | 0.9951 | 0.83 | 0.9847 | 1.34 |
| | RF-SPAD | RSI764,766—RSI768,770—RSI772,762—RSI$_{766,764}$ | 0.9985 | 0.25 | 0.9826 | 1.53 |
| | DT-SPAD | RSI764,766—RSI768,770—RSI772,762—RSI$_{766,764}$ | 0.9999 | 0.02 | 0.9756 | 2.14 |
| Firmness | ANN-Firmness | RSI764,766—RSI768,770—RSI772,762—RSI766,764—RSI760,858—RSI$_{970,1000}$ | 0.9925 | 1.28 | 0.9814 | 1.22 |
| | RF- Firmness | RSI764,766—RSI768,770—RSI772,762—RSI766,764—RSI760,858—RSI$_{970,1000}$ | 0.9972 | 0.49 | 0.9909 | 0.59 |
| | DT- Firmness | RSI764,766—RSI768,770—RSI772,762—RSI766,764—RSI760,858—RSI$_{970,1000}$ | 0.9998 | 0.03 | 0.9853 | 0.96 |

The complete names of the abbreviated hyperspectral indices can be found in Table 1.

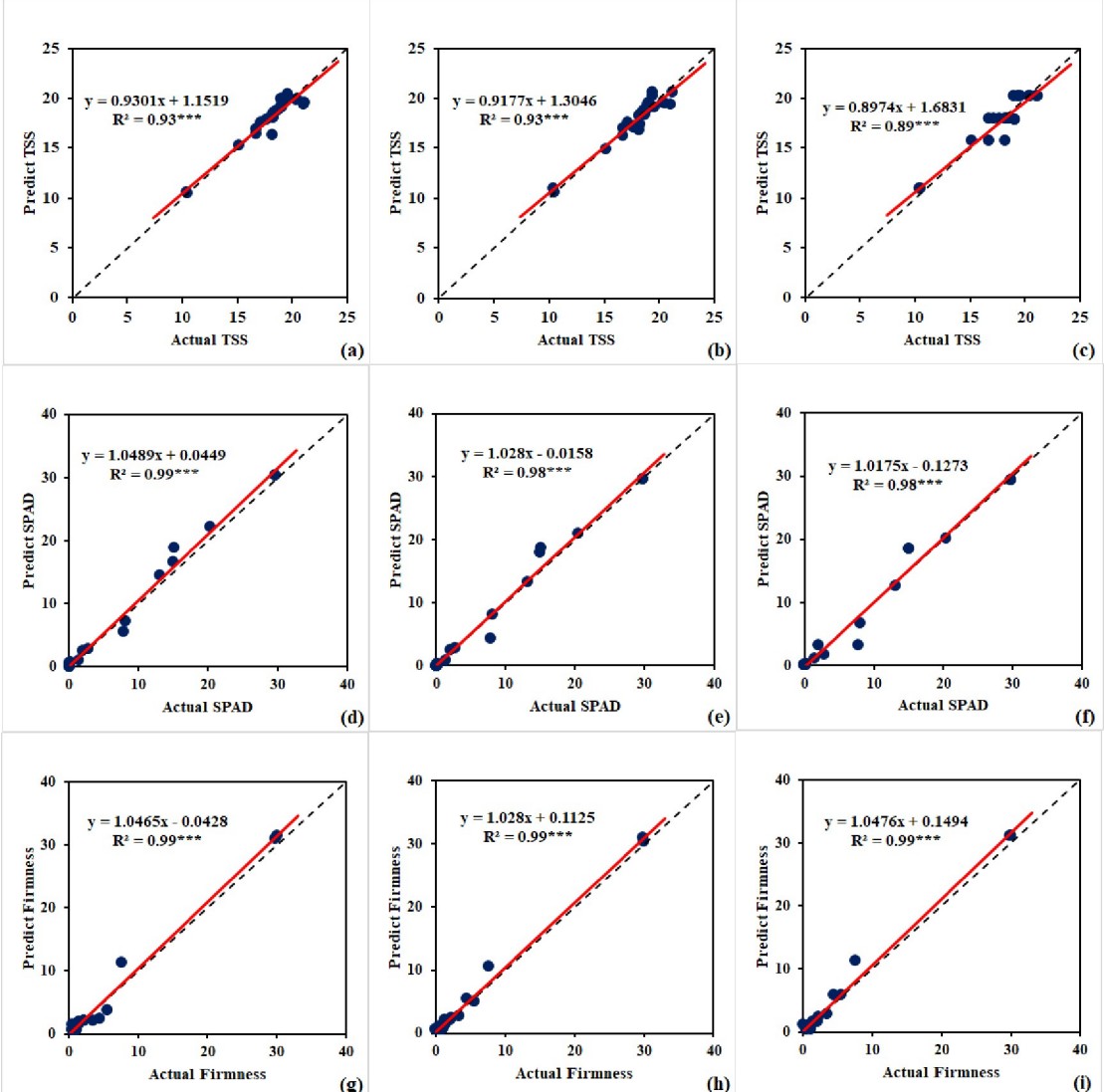

**Fig 7.** Comparative Analysis of (a, d, g) artificial neural network, (b, e, h) random forest, and (c, f, i) decision trees for total soluble solids (TSS), SPAD, and firmness for mango during testing phase.

employed the squared error as the criterion function. In contrast, the decision tree (DT-SPAD) model exhibited an $R^2$ value of 0.99 and 0.98, accompanied by MSE values of 0.02 and 2.14 for the training and testing phases, respectively. Constructed with a maximum depth of 5 and utilizing the squared error as the criterion function. These impressive results further emphasize the efficacy of these machine learning models in forecasting SPAD values, underscoring their significance in precise mango quality assessment.

In terms of firmness prediction, the artificial neural network (ANN-Firmness) model performed exceptionally well, with $R^2$ values of 0.99 and 0.98 and matching MSE values of 1.28 and 1.22 for the training and testing phases, respectively. Four neurons, each in two hidden layers, were used to construct the ANN- Firmness (Fig 6C). This model used the relu activation function. Its performance was optimized through 700 iterations. The random forest (RF-Firmness) model provides accuracy in predicting mango fruit's firmness, with an astounding $R^2$ value of 0.99 and MSE values of 0.49 and 0.59 for the training and testing phases, respectively.

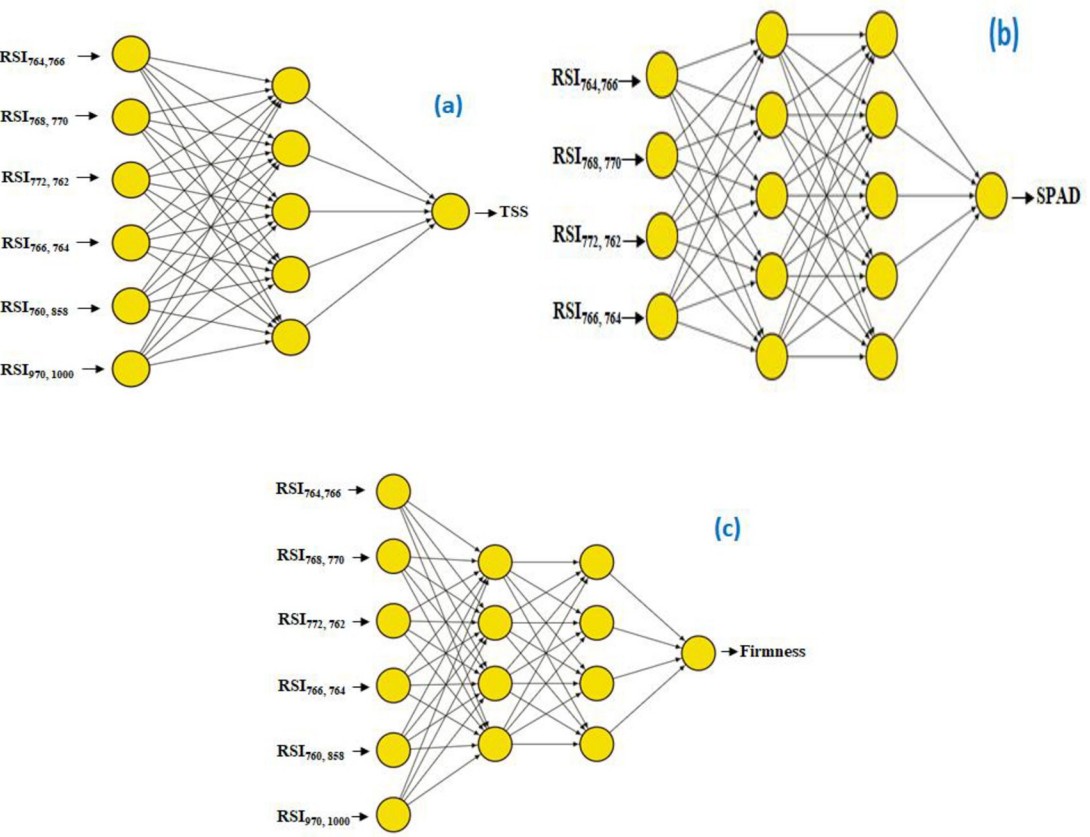

**Fig 8. ANN architecture, incorporating a combination of the effective spectral reflectance indices, for identifying total soluble solids (TSS), SPAD, and firmness of mango fruits.**

The RF-Firmness model consisted of 20 trees with a maximum depth of 5. The squared error served as the criteria function for this model. With an $R^2$ value of 0.99 and 0.99 and matching MSE values of 0.03 and 0.96, throughout the training and testing stages, respectively. The decision tree (DT- Firmness) model demonstrated exceptional performance. With a maximum depth of 5 and the squared error served as the criterion function. These outcomes show how well machine learning models predict the mango fruit's firmness.

### 3.6. ML-models performance for strawberry quality assessment

Table 8 and Fig 9 illustrate how well ML models have predicted quality parameters for strawberries. For the L* prediction, the artificial neural network (ANN-L*) model demonstrated remarkable accuracy with $R^2$ values of 0.87 and 0.75 and corresponding MSE values of 2.20 and 3.07, during the training and testing phases, respectively. As seen in Fig 10A, the ANN-L* model was created with two hidden layers, each containing four neurons. It ran through 500 iterations using the relu activation function. The random forest (RF-L*) model achieved high accuracy, obtaining $R^2$ values of 0.88 and 0.74 and MSE values of 1.84 and 3.17 for the training and testing phases, respectively. This model used squared error as a criterion function. It consists of 15 trees with a maximum depth of 2. Furthermore, during the training and testing stages, the decision tree (DT-L*) model showed excellent performance, with $R^2$ values of 0.91 and 0.74 and MSE values of 1.52 and 3.27, respectively. This model was built with a maximum depth of 2. It used the squared error as the criteria function. These results confirm the models'

**Table 8. Performance of different ML models incorporate distinct spectral reflectance indices (SRIs) to predict L\*, b\*, total soluble solids (TSS), and firmness of Strawberry after training and testing phase.**

| Target Variables | Models | Optimal Input Variables | Training | | Testing | |
|---|---|---|---|---|---|---|
| | | | $R^2$ | MSE | $R^2$ | MSE |
| L* | ANN-L* | RSI666,636—RSI660,620—RSI670,610—RSI618,602—RSI640,590—RSI970,590 –GI–PRMI–NDVI$_{780, 670}$ | 0.8659 | 2.20 | 0.7525 | 3.07 |
| | RF-L* | RSI666,636—RSI660,620—RSI670,610—RSI618,602—RSI640,590—RSI970,590—GI | 0.8879 | 1.84 | 0.7443 | 3.17 |
| | DT-L* | RSI666,636—RSI660,620—RSI670,610—RSI618,602—RSI640,590—RSI970,590—GI | 0.9073 | 1.52 | 0.7360 | 3.27 |
| b | ANN-b* | RSI666,636—RSI660,620—RSI670,610—RSI618,602—RSI640,590—RSI970,590—GI | 0.9139 | 1.43 | 0.9102 | 1.38 |
| | RF-b* | RSI666,636—RSI660,620—RSI670,610—RSI618,602—RSI640,590—RSI970,590 –GI–PRMI–NDVI$_{780, 670}$ | 0.9148 | 1.42 | 0.8438 | 2.40 |
| | DT-b* | RSI666,636—RSI660,620—RSI670,610—RSI618,602—RSI640,590—RSI970,590—GI | 0.7787 | 3.68 | 0.7942 | 3.16 |
| TSS | ANN-TSS | RSI666,636—RSI660,620—RSI670,610—RSI618,602—RSI640,590—RSI970,590 –GI–PRMI–NDVI$_{780, 670}$ | 0.7541 | 0.20 | 0.5836 | 0.37 |
| | RF-TSS | RSI666,636—RSI660,620—RSI670,610—RSI618,602—RSI640,590—RSI970,590 –GI–PRMI–NAI—NDVI$_{780, 670}$ | 0.8684 | 0.11 | 0.8568 | 0.13 |
| | DT-TSS | RSI666,636—RSI660,620—RSI670,610—RSI618,602—RSI640,590—RSI970,590 –GI–PRMI–NDVI$_{780, 670}$ | 0.7528 | 0.21 | 0.8002 | 0.18 |
| Firmness | ANN-Firmness | RSI666,636—RSI660,620—RSI670,610—RSI618,602—RSI640,590—RSI970,590 –GI–PRMI–NDVI$_{780, 670}$ | 0.8135 | 0.39 | 0.839 | 0.36 |
| | RF- Firmness | RSI666,636—RSI660,620—RSI670,610—RSI618,602—RSI640,590—RSI970,590—GI | 0.8532 | 0.31 | 0.8004 | 0.44 |
| | DT- Firmness | RSI666,636—RSI660,620—RSI$_{670,610}$ | 0.8609 | 0.29 | 0.8083 | 0.43 |

The complete names of the abbreviated hyperspectral indices can be found in Table 1.

capacity to accurately predict L\*. For b\* prediction, the artificial neural network (ANN-b\*) model exhibited exceptional performance, with $R^2$ values of 0.91 and 0.91 for the training and testing phases, respectively. The MSE values that were consistent were 1.43 and 1.38. The ANN-b\* model was developed using one hidden layer, each of which contained seven neurons (Fig 10B). The relay activation function was employed to optimize its performance over 700 iterations. In the same vein, the random forest (RF-b\*) model demonstrated exceptional accuracy, achieving $R^2$ values of 0.91 and 0.84 for the training and assessment phases, respectively, with MSE values of 1.42 and 2.40. The RF-b\* model consisted of 10 trees with a maximum depth of 2. It employs the squared error as the criterion function. In contrast, the decision tree (DT-b\*) model achieved an $R^2$ value of 0.78 and 0.79, accompanied by MSE values of 3.68 and 3.16 during the training and testing phases, respectively. The DT-b\* model was built with a maximum depth of 1, and the squared error was used as the criterion function. These remarkable outcomes demonstrate the effectiveness of these machine learning models in precisely predicting b\* values. These ML models underline their importance in accurately predicting the quality of strawberry fruit. For TSS prediction, the artificial neural network (ANN-TSS) model performed exceptionally well, with $R^2$ values of 0.76 and 0.58 and MSE values of 0.20 and 0.37, for the training and testing phases, respectively. Five neurons make up its single hidden layer architecture (Fig 10C). The Tanh activation function was employed. For performance optimization, it went through 500 iterations. Likewise, the random forest (RF-TSS) model demonstrated excellent accuracy, achieving training and testing phases' $R^2$ values of 0.87 and 0.86 along with MSE values of 0.11 and 0.13. The RF-TSS model is made up of 20 trees with a maximum depth of 5. It used the squared error as the criteria function. On the other hand, throughout the training and testing stages, the decision tree (DT-TSS) model obtained an $R^2$ value of 0.75 and 0.80, with MSE values of 0.21 and 0.18, respectively. Using the squared error as the criterion function. The DT-TSS model was built with a maximum

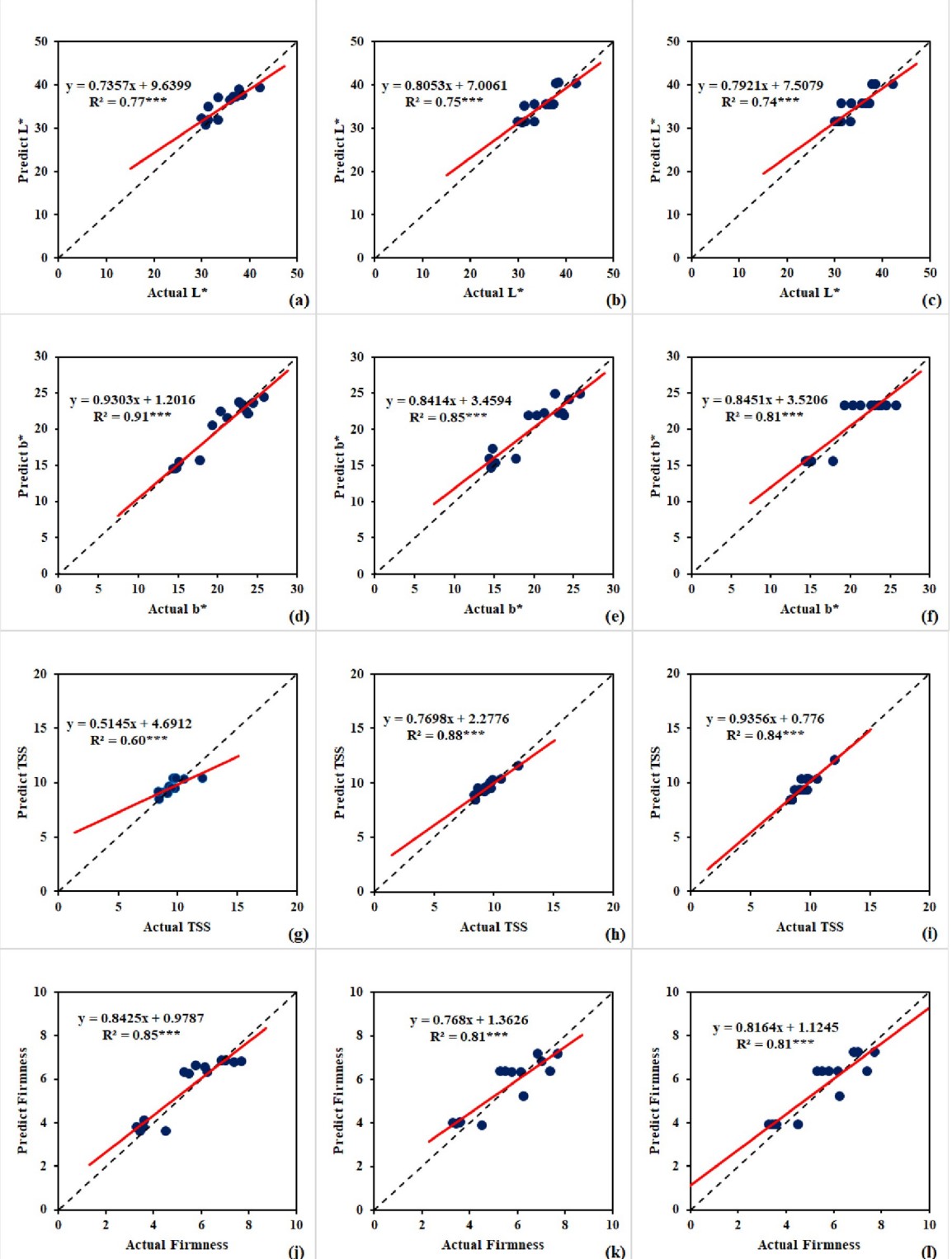

**Fig 9.** Comparative Analysis of (a, d, g, j) Artificial Neural Network, (b, e, h, k) Random Forest, and (c, f, i, l) Decision Trees for L*, b*, total soluble solids (TSS), and firmness for strawberry during testing phase.

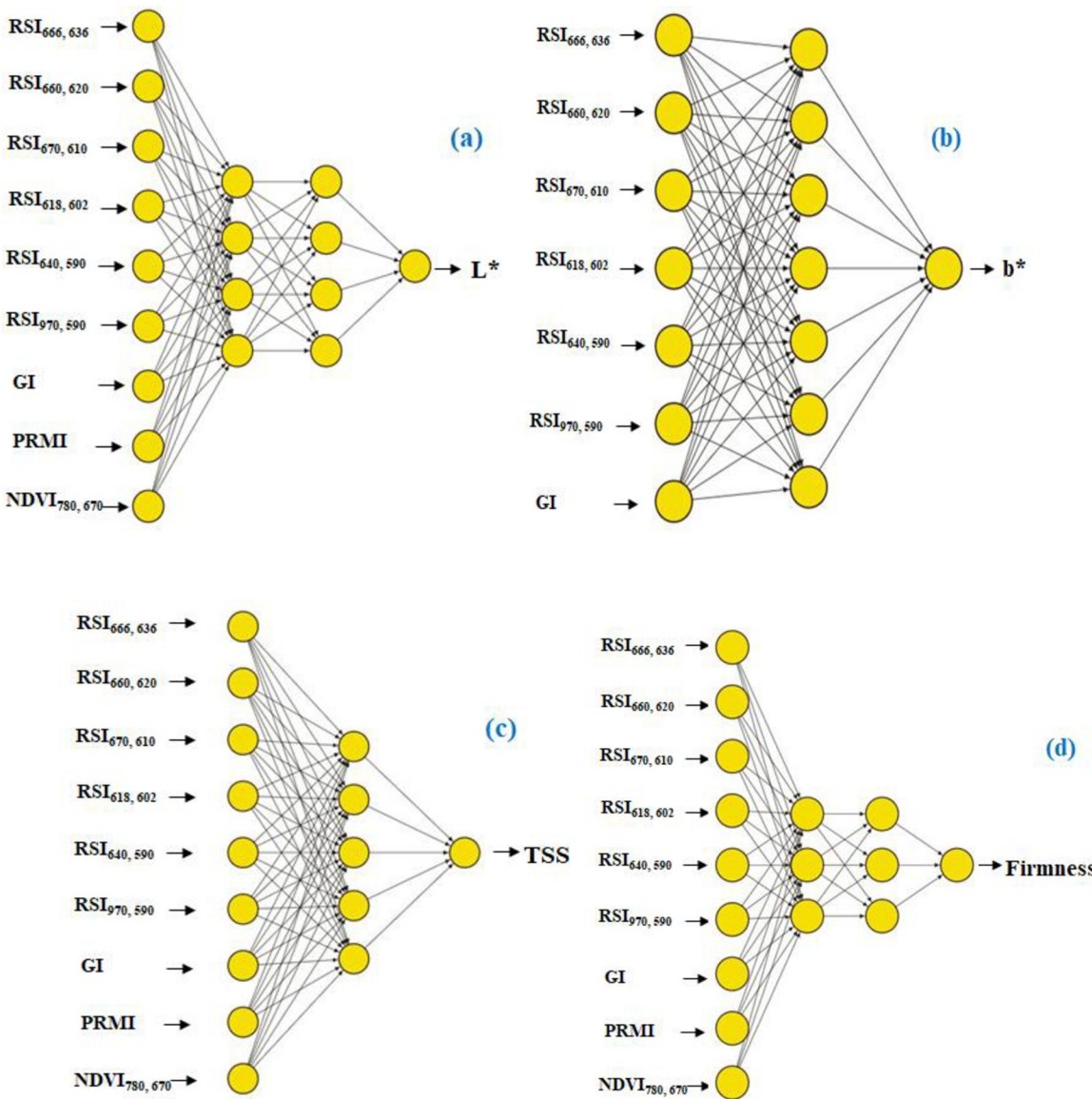

**Fig 10. ANN architecture, incorporating a combination of the effective spectral reflectance indices, for identifying L\*, b\*, total soluble solids (TSS), and firmness of strawberry fruits.**

depth of 2. These remarkable findings demonstrate the effectiveness of these machine learning models in precisely predicting TSS levels. In terms of firmness prediction, the artificial neural network (ANN-Firmness) model performed exceptionally well, showing $R^2$ values of 0.81 and 0.84, respectively, for training and testing phases. According to the corresponding MSE values of 0.39 and 0.36, the model can accurately measure firmness. The ANN-Firmness model was built using two hidden layers, each with three neurons (Fig 10D). Tanh activation function was employed. Its performance was optimized through 700 iterations. In a similar vein, the random forest (RF-Firmness) model showed excellent accuracy, with $R^2$ values for the training and testing phases of 0.85 and 0.80, respectively, and MSE values of 0.31 and 0.44. The RF-Firmness model is made up 20 trees with a maximum depth of 2. It used the squared error as the criteria function. Finally, the decision tree (DT-Firmness) model achieved $R^2$ with 0.86

and 0.80, the corresponding MSE values of 0.29 and 0.81, for the training and testing phases, respectively. The DT- Firmness model has a maximum depth of 2. The criteria function for this model was the squared error. These remarkable results show how accurate and reliable these ML models are at predicting firmness. Their ability to generate reliable judgments emphasizes their significance in precisely evaluating strawberry quality.

The traditional biochemical measurements are based on chemical analysis techniques in the laboratory, which, making it impossible to obtain information on biochemical measurements in a timely manner and on a large scale. Determining quality parameters in response to different levels of ripening is beneficial for both harvest and export. The use reflectance measurements could help to obtain data on various types of fruits at different ripening stages at a low cost. Developing an active spectral device that incorporates the specific wavelengths identified in the study, instead of the passive spectral device used in our study, helps to avoid the limitations posed by dependency on sunlight. This extension allows for its application in interior environments, such as manufacturing lines and factories. This innovation has the potential to enhance the usefulness and application of spectral analysis tools in industrial and research contexts by enabling operation at any time and in various environments. Machine vision-based reflectance measurements do not directly furnish details about fruit quality parameters necessitating the use of spectral reflectance indices (SRIs). SRIs are formulated by combining data from multiple spectral wavelengths unaffected by environmental conditions, thereby offering more efficient insights about fruit quality parameters. Once relevant SRIs are established, simpler sensors that measure specific spectral wavelengths can be deployed in real-time, presenting a cost-efficient strategy. Additional, to counteract the adverse impacts of environmental conditions, the amalgamation of diverse SRIs into a unified index proves beneficial. In our study, different SRIs were integrated into ML models, leveraging its correlation above a predefined threshold to predict quality parameters for mango and strawberry fruits. Employing a combination of SRIs has been shown to enhance the indirect assessment of quality parameters for mango and strawberry fruits. Tables 7 and 8 delineate the implementation of ANN, RF, and DT models which incorporate the analysis of high-level variables using a combination of spectral reflectance indices (SRIs). This process aids in the detection of three parameters (SPAD value, TSS, and firmness) of mango fruits, as well as four parameters ($L^*$, $b^*$, TSS, and firmness) of strawberry fruits across the ripening stages. The ML models were trained using the SRIs to predict the examined parameters, with the actual values of the ML model being compared against the predicted values. The study's multivariate analysis and comparison techniques indicate a significant increase in predictability when applying this approach. These findings are consistent with previous studies, Mithun et al. [91] used spectral reflectance data points taken from the spectral range of 350 nm to 1050 nm, combined with RF and DT models, for the detection of artificially ripened mango. Their results exhibited high accuracy rates of 97.22% for RF and 98.91% for DT. Munawar [92] reported that the optimal models for mango quality parameter prediction were achieved by coupling ANN with NIR. Zhang et al. [93] employed NIR data collected during the mango ripening season incorporated with an ANN model for a mango fruit's TSS detection, resulting in a $R^2$ of 0.9055 and an RMSE of 0.2192. Devassy and George [52] showed the effectiveness of using spectral reflectance with an RF model to predict the strawberry fruit's firmness in a rapid, non-destructive method with an $R^2$ of prediction of 0.90 and 0.94 for training and testing, respectively. Amoriello et al. [94] investigated the feasibility of employing ANN models in conjunction with spectral reflectance in the wavelength range of 400–1000 nm to forecast strawberry quality parameters. Prediction models were observed with an $R^2 = 0.81$ for firmness, 0.96 for TSS, 0.88 for titratable acidity, and 0.95 for dry matter. ElMasry et al. [95] used color features extracted from strawberry

images, which contained the brightness values, for fruit firmness prediction using ANN. The efficiency of the model in classifying fruits was 92.88%.

## 4. Future outlook

The research should be extended to suggest development of an active spectral device that incorporates specific wavelengths to determine quality parameters. This initiative introduces a novel technological solution that could revolutionize the evaluation process for fruits. The application of ML models in fruit quality control could be expanded by adapting the successful methods for evaluating the quality of strawberries and mangoes to other fruits or various types. The development of user-friendly graphical interfaces for farmers will be critical to the adoption of these technologies. Longitudinal studies that assess fruit quality over time will provide insights into how environmental factors influence fruit quality. This approach can lead to more robust predictive models.

## 5. Conclusions

This study was to evaluate the biochemical parameters of mango and strawberry fruits at various ripening stages. This was done by utilizing a combination of established and newly developed SRIs in conjunction with ML models, including ANN, RF, and DT. Our results revealed significant differences in the fruit's biochemical parameters across various ripening stages. The published SRIs showed strong correlations with the selected parameters, but the newly developed SRIs were more effective at monitoring the various biochemical parameters. The integration of SRIs with diverse ML models proved to be a successful strategy for precisely estimating biochemical parameters. For mango biochemical parameters prediction, the ANN models demonstrated $R^2$ values ranging from 0.92 to 1.00 and from 0.93 to 0.98 for training and testing, respectively. On the other hand, the RF models exhibited $R^2$ values ranging from 0.98 to 1.00 and from 0.93 to 0.99 during training and testing, respectively. The DT models showed highly performance with $R^2$ values ranging from 0.95 to 1.00 and 0.88 to 0.99 for training and testing phases. For strawberry's biochemical parameters prediction, the ANN models achieved $R^2$ values between 0.75 and 0.91 and between 0.58 and 0.91 during training and testing phases, respectively. On the other hand, RF models showed $R^2$ values between 0.85 and 0.91 during training and between 0.74 and 0.86 during testing. The DT models demonstrated good results, with $R^2$ values ranging from 0.75 to 0.91 for the training set and 0.74 to 0.81 for the testing set. These findings underscore the effectiveness of combining SRIs with ML models, particularly ANN, RF, and DT, for accurate prediction of biochemical parameters in mango and strawberry fruits.

## Author Contributions

**Conceptualization:** Salah Elsayed.

**Data curation:** Salah Elsayed, Hoda Gala, Mohamed S. Abd El-baki, Ahmed Elbeltagi, Nadia G. Abd El-Fattah.

**Formal analysis:** Salah Elsayed, Hoda Gala, Mohamed S. Abd El-baki, Mohamed Maher, Ahmed Elbeltagi, Osama Elsherbiny, Nadia G. Abd El-Fattah.

**Funding acquisition:** Ahmed Elbeltagi, Ali Salem.

**Investigation:** Hoda Gala, Mohamed Maher, Ahmed Elbeltagi, Abdallah Elshawadfy Elwakeel.

**Methodology:** Salah Elsayed, Mohamed Maher, Ahmed Elbeltagi, Osama Elsherbiny, Nadia G. Abd El-Fattah.

**Project administration:** Hoda Gala, Mohamed Maher, Ahmed Elbeltagi, Ali Salem.

**Resources:** Mohamed Maher.

**Software:** Mohamed S. Abd El-baki, Nadia G. Abd El-Fattah.

**Supervision:** Salah Elsayed, Mohamed Maher, Ahmed Elbeltagi, Ali Salem.

**Validation:** Mohamed S. Abd El-baki, Abdallah Elshawadfy Elwakeel, Nadia G. Abd El-Fattah.

**Writing – original draft:** Salah Elsayed, Hoda Gala, Mohamed S. Abd El-baki, Abdallah Elsha-wadfy Elwakeel, Nadia G. Abd El-Fattah.

**Writing – review & editing:** Hoda Gala, Ahmed Elbeltagi, Ali Salem, Abdallah Elshawadfy Elwakeel, Osama Elsherbiny, Nadia G. Abd El-Fattah.

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
