## [Decision Letter · Decision Letter 0]

13 Sep 2024

PONE-D-24-31212Hyperspectral Technology and Machine Learning Models to Estimate the Fruit Quality Parameters of Mango and Strawberry CropsPLOS ONE

Dear Dr. Elbeltagi,

Thank you for submitting your manuscript to PLOS ONE. After careful consideration, we feel that it has merit but does not fully meet PLOS ONE’s publication criteria as it currently stands. Therefore, we invite you to submit a revised version of the manuscript that addresses the points raised during the review process.

We look forward to receiving your revised manuscript.

Kind regards,

**Sajid Ali**

Academic Editor

PLOS ONE

Journal Requirements:

1. When submitting your revision, we need you to address these additional requirements. Please ensure that your manuscript meets PLOS ONE's style requirements, including those for file naming. The PLOS ONE style templates can be found at https://journals.plos.org/plosone/s/file?id=wjVg/PLOSOne_formatting_sample_main_body.pdf and https://journals.plos.org/plosone/s/file?id=ba62/PLOSOne_formatting_sample_title_authors_affiliations.pdf 2. In your Methods section, please provide additional information regarding the permits you obtained for the work. Please ensure you have included the full name of the authority that approved the field site access and, if no permits were required, a brief statement explaining why. 3. Please note that PLOS ONE has specific guidelines on code sharing for submissions in which author-generated code underpins the findings in the manuscript. In these cases, we expect all author-generated code to be made available without restrictions upon publication of the work. Please review our guidelines at https://journals.plos.org/plosone/s/materials-and-software-sharing#loc-sharing-code and ensure that your code is shared in a way that follows best practice and facilitates reproducibility and reuse. 4. In the online submission form, you indicated that "Data will be available upon request" All PLOS journals now require all data underlying the findings described in their manuscript to be freely available to other researchers, either 1. In a public repository, 2. Within the manuscript itself, or 3. Uploaded as supplementary information.This policy applies to all data except where public deposition would breach compliance with the protocol approved by your research ethics board. If your data cannot be made publicly available for ethical or legal reasons (e.g., public availability would compromise patient privacy), please explain your reasons on resubmission and your exemption request will be escalated for approval.

Reviewers' comments:

Reviewer's Responses to Questions

**Comments to the Author**

1. Is the manuscript technically sound, and do the data support the conclusions?

Reviewer #1: Yes

Reviewer #2: Yes

2. Has the statistical analysis been performed appropriately and rigorously? 

Reviewer #1: Yes

Reviewer #2: Yes

3. Have the authors made all data underlying the findings in their manuscript fully available?

Reviewer #1: No

Reviewer #2: Yes

4. Is the manuscript presented in an intelligible fashion and written in standard English?

Reviewer #1: No

Reviewer #2: Yes

5. Review Comments to the Author

**Reviewer #1:** 1- Introduction suffers from lack of motivations and innovations. It should be expanded to include a more detailed discussion of current problems. Use the following articles to improve the introduction section and other sections:

https://doi.org/10.1016/j.chemolab.2022.104650

https://doi.org/10.1016/j.jafr.2023.100931

https://doi.org/10.1016/j.ecoinf.2022.101829

2- What is the difference between evaluation of fruit ripening level using conventional image processing and hyperspectral technology?

3- A section on network training should be included which details on the parameters (e.g., learning rate, number of iterations, batch size, momentum etc.) used for training algorithms (what filter size?). what is the architecture of algorithms? Such information would increase the quality of manuscript and also provides a way for re-implementation by others.

4- Actual views of the equipment used and all samples should be provided in the manuscript.

5- Results and Discussion; author should compare the finding of present study with previous study and justify for more clarity.

6- The captions of some figures and tables are not descriptive enough.

7- The technical discussion is insufficient, requiring a more thorough analysis and interpretation of results to substantiate the scientific merit of the study.

8- Author should add separate section regarding future outlook and specific comment point wise based on their study.

9- Limitations of the study should be included and discussed.

**Reviewer #2:** The manuscript is interesting providing a suitable method to evaluate the fruit ripening. The manuscript is well-designed and will-written. only some minor comments are there in the attached manuscript need to be addressed before acceptance in order to improve the manuscript for publication

6. PLOS authors have the option to publish the peer review history of their article (what does this mean?). If published, this will include your full peer review and any attached files.

Reviewer #1: No

Reviewer #2: No

---

## [Author Response · Author response to Decision Letter 0]

2 Oct 2024

Reviewer 1 

1 Introduction suffers from lack of motivations and innovations. It should be expanded to include a more detailed discussion of current problems. Use the following articles to improve the introduction section and other sections:

https://doi.org/10.1016/j.chemolab.2022.104650

https://doi.org/10.1016/j.jafr.2023.100931

https://doi.org/10.1016/j.ecoinf.2022.101829

Response: Thank you for your comments. We have improved the introduction section and other sections using these articles, as you requested.

2 What is the difference between evaluation of fruit ripening level using conventional image processing and hyperspectral technology? 

Conventional Image Processing: Uses visible light images (RGB), analyzes color, shape, and texture, Limited to surface characteristics. less accurate for subtle changes in ripeness. Struggling with varying lighting conditions.

Hyperspectral Technology: Captures a wide range of wavelengths. Provides detailed chemical composition and internal properties. Can detect subtle differences in ripening stages based on spectral signatures. Offers higher accuracy in determining ripeness levels, including internal quality indicators. Better at identifying diseases or defects not visible in conventional images.

3 A section on network training should be included which details on the parameters (e.g., learning rate, number of iterations, batch size, momentum etc.) used for training algorithms (what filter size?). what is the architecture of algorithms? Such information would increase the quality of manuscript and also provides a way for re-implementation by others. 

Thank you for your comments. We have included additional details for the parameters used for training algorithms in the Materials and Methods section. This information can be found in lines 223-226 for artificial neural network model, lines 235-238 for random forest model, and lines 252-255 for decision tree model. Additionally, we have already incorporated information in the Results and Discussion section. Specifically, this information can be found in lines 412-447 for the mango’s models and lines 453-499 for the strawberry’s models, emphasizing the optimal hyperparameters utilized in our models.

4 Actual views of the equipment used and all samples should be provided in the manuscript. 

Thank you for your comments. We have provided figures for the equipment used in our manuscript, as you requested.

5 Results and Discussion; author should compare the finding of present study with previous study and justify for more clarity. 

We have already added recent studies in lines (273-303), (306-337), (353-372), and (384-413) for quality parameters and spectral reflectance indices and in lines (412-447) and (453-499) for our models.

6 The captions of some figures and tables are not descriptive enough. 

Thank you for your comments. We have ensured that all captions of figures and tables are descriptive enough, as you requested.

7 The technical discussion is insufficient, requiring a more thorough analysis and interpretation of results to substantiate the scientific merit of the study. 

Thank you for your comments. We have already added information in lines (505-544)

8 Author should add separate section regarding future outlook and specific comment point wise based on their study. 

Thank you for your comments. We have added section regarding future outlook in line (545-553), as you requested.

9 Limitations of the study should be included and discussed. 

Thank you for your comments. We have included limitations of the study in discussion in line (509-511).

Reviewer 2

1 The manuscript is interesting providing a suitable method to evaluate the fruit ripening. The manuscript is well-designed and will-written. only some minor comments are there in the attached manuscript need to be addressed before acceptance in order to improve the manuscript for publication.

Thank you for comment. We have improved our manuscript according to the comments mentioned. Thank you for your special addition.

---

## [Decision Letter · Decision Letter 1]

24 Oct 2024

Hyperspectral Technology and Machine Learning Models to Estimate the Fruit Quality Parameters of Mango and Strawberry Crops

PONE-D-24-31212R1

Dear Dr. Salem,

We’re pleased to inform you that your manuscript has been judged scientifically suitable for publication and will be formally accepted for publication once it meets all outstanding technical requirements.

Kind regards,

**Sajid Ali**

Academic Editor

PLOS ONE

Additional Editor Comments (optional):

Reviewers' comments:

Reviewer's Responses to Questions

**Comments to the Author**

1. If the authors have adequately addressed your comments raised in a previous round of review and you feel that this manuscript is now acceptable for publication, you may indicate that here to bypass the “Comments to the Author” section, enter your conflict of interest statement in the “Confidential to Editor” section, and submit your "Accept" recommendation.

Reviewer #1: All comments have been addressed

Reviewer #2: All comments have been addressed

2. Is the manuscript technically sound, and do the data support the conclusions?

Reviewer #1: Yes

Reviewer #2: Yes

3. Has the statistical analysis been performed appropriately and rigorously? 

Reviewer #1: Yes

Reviewer #2: Yes

4. Have the authors made all data underlying the findings in their manuscript fully available?

Reviewer #1: Yes

Reviewer #2: Yes

5. Is the manuscript presented in an intelligible fashion and written in standard English?

Reviewer #1: Yes

Reviewer #2: Yes

6. Review Comments to the Author

Reviewer #1: The authors have significantly improved the quality of the manuscript and the presentation of results. They have satisfactorily attended to all the reviewer's comments. Hence the manuscript can be accepted.

Reviewer #2: The author responded to all comments. the manuscript can be accepted if there is no negative comments from other reviewers

7. PLOS authors have the option to publish the peer review history of their article (what does this mean?). If published, this will include your full peer review and any attached files.

Reviewer #1: No

Reviewer #2: No

---

## [Editor Report · Acceptance letter]

1 Nov 2024

PONE-D-24-31212R1 

PLOS ONE

Dear Dr. Salem, 

I'm pleased to inform you that your manuscript has been deemed suitable for publication in PLOS ONE. Congratulations! Your manuscript is now being handed over to our production team.

Kind regards, 

on behalf of

Dr. Sajid Ali 

Academic Editor

PLOS ONE